# Associations of clothing size, adiposity and weight change with risk of postmenopausal breast cancer in the UK Women's Cohort Study (UKWCS)

Foong Ming Moy,[1,2] Darren C Greenwood,[3] Janet E Cade[2]

[1]Department of Social & Preventive Medicine, Faculty of Medicine, University of Malaya, Kuala Lumpur, Malaysia
[2]Nutritional Epidemiology Group, School of Food Science & Nutrition, University of Leeds, Leeds, UK
[3]Division of Epidemiology & Biostatistics, School of Medicine, University of Leeds, Leeds, UK

**Correspondence to**
Dr Foong Ming Moy;
moyfm@ummc.edu.my

## ABSTRACT

**Objectives** Breast cancer is associated with overweight and obesity after menopause. However, clothing size as a proxy of adiposity in predicting postmenopausal breast cancer is not widely studied. We aimed to explore the relationships between postmenopausal breast cancer risk with adipose indicators (including clothing sizes) and weight change over adulthood.

**Design** Prospective cohort study.

**Setting** England, Wales and Scotland.

**Participants** 17781 postmenopausal women from the UK Women's Cohort Study.

**Primary outcome measure** Incident cases of malignant breast cancers (International Classification of Diseases (ICD) 9 code 174 and ICD 10 code C50).

**Results** From 282277 person-years follow-up, there were 946 incident breast cancer cases with an incidence rate of 3.35 per 1000 women. Body mass index (HR: 1.04; 95%CI: 1.02 to 1.07), blouse size (HR: 1.10; 1.03 to 1.18), waist circumference (HR: 1.07; 1.01 to 1.14) and skirt size (HR: 1.14;1.06 to 1.22) had positive associations with postmenopausal breast cancer after adjustment for potential confounders. Increased weight over adulthood (HR: 1.02; 1.01 to 1.03) was also associated with increased risk for postmenopausal breast cancer.

**Conclusions** Blouse and skirt sizes can be used as adipose indicators in predicting postmenopausal breast cancer. Maintaining healthy body weight over adulthood is an effective measure in the prevention of postmenopausal breast cancer.

## INTRODUCTION

Breast cancer is the most common cancer among women worldwide, accounting for 25% of new cases.[1] In the UK, it is also the most common cancer among women with around 54800 cases diagnosed in 2014.[2] The prevalence of obesity in the UK has increased from 15% in 1993 to 27% in 2015, and 57% of the women were overweight or obese.[3] Obesity is associated with increased risks of chronic diseases and certain cancers.[4 5] There is strong evidence that obesity and weight gain over adult life are associated with increased risk of postmenopausal breast

### Strengths and limitations of this study

► This study may be the first to report the use of both blouse and skirt sizes as measures of body size.
► The large study sample with long follow-up duration and linkage of participants' ID with the National Statistics National Health Service central register provides accurate information on breast cancer diagnosis.
► All adipose indicators, blouse and skirt sizes, were self-reported.
► We were unable to evaluate in more detail the effect of weight gain at different ages.
► Breast tumour characteristics were not available, and we could not explore whether our findings may differ by breast cancer subtypes.

cancer.[6 7] Maintaining healthy weight is a key recommendation to prevent postmenopausal breast cancer.[6]

To date, many adipose indicators namely weight, body mass index (BMI), waist circumference, waist to hip ratio are used in research and health education programmes. BMI is an established indicator for general obesity, while central obesity is commonly measured using waist circumference. Among postmenopausal women, higher BMI and larger waist circumference during adulthood, and weight gain over adulthood are associated with increased breast cancer risk.[6]

Clothing sizes are objective, easy to recall and less likely to be misreported. From a public health perspective, clothing sizes may be useful when anthropometric measurements are not available or challenging in their measurements due to extreme obesity. Skirt or trouser size[8–10] have been suggested to be used as proxy for central obesity. Skirt size appears to predict cancer risk independently of BMI.[10] Change in skirt size was also found to be predictive of postmenopausal breast cancer.[11] However, blouse size has not been

explored as a proxy for general obesity in predicting postmenopausal breast cancer. In view of the limited evidence of using clothing size as an adiposity indicator in predicting postmenopausal breast cancer risk, and the ease of self-reporting of these measures, we have used data from the UK Women's Cohort Study (UKWCS) to explore the relationships between postmenopausal breast cancer risk with various adipose indicators (including skirt and blouse sizes); as well as weight change over adulthood.

## MATERIALS AND METHODS
### Study design and population
The recruitment of participants and characteristics of the UKWCS have been described in detail previously.[12] In brief, a total of 35 792 women were recruited between 1995 and 1998 via a postal questionnaire. They were aged 35–69 years at baseline and were living in England, Wales and Scotland. In total, 17 781 women were postmenopausal, and 15 951 women were premenopausal at baseline. Postmenopausal women were those who had not had a natural menstrual period in the last 12 months, or were older than 50 years and currently on hormone replacement therapy (HRT), or with a previous hysterectomy and HRT. Where these data were missing, if the woman was aged more than 50 years she was also counted as menopausal. Only postmenopausal women were included in the current analyses.

### Ethics clearance
A total of 174 local research ethics committees were contacted and permission to carry out the baseline UKWCS study was obtained.[13] Participants had consented to the use of information gathered at baseline, future phases and cancer registries for research purposes provided that confidentiality was maintained. The National Research Ethics Committee for Yorkshire and the Humber, Leeds East have now taken on responsibility for the on-going cohort.[14]

### Patient and public involvement
The participants were not involved in the development of the research question and outcome measures. They were also not involved in the design and conduct of the study. The results are not disseminated to the participants apart from the UKWCS website.

### Case definition and ascertainment
All women were flagged for death and cancer registration on the office of National Statistics National Health Service central register. Causes of death and cancer diagnosis were coded in the International Classification of Diseases (ICD) 9 and 10. All malignant breast cancers (ICD 9 code 174 and ICD 10 code C50) registered after the participants returned their questionnaires were taken as incident cases. Prevalent cancers identified from the questionnaires and incident cases identified within 1 year from the date of questionnaire returned were excluded

to rule out reverse causation since undiagnosed breast cancer may influence weight.

### Assessment of adipose indicators
Anthropometric measures such as weight, height, waist and hip circumference at enrolment; and weight at age 20 years were self-recorded. BMI was categorised into groups of underweight, normal weight, overweight and obese according to the WHO criteria.[15] Weight change over adulthood was computed by weight at enrolment minus weight at 20 years old. Weight change was further categorised into stable weight (±2 kg), weight loss (2–4.9 kg and >5 kg) and weight gain (2–9.9 kg, 10–19.9 kg and >20 kg). Skirt size and blouse size at enrolment were also reported. The clothing sizes ranged from 6 to 26 with the increase in units of 2.

### Other confounding factors
Socioeconomic characteristics, dietary pattern (meat eaters, fish eaters or vegetarian), lifestyle (smoking, alcohol consumption, physical activity, etc), reproductive history, information on past health experience, sibling and parental health were reported at baseline.

### Statistical analysis
The follow-up (person-years) for each participant was counted from the beginning of the study until the date of breast cancer diagnosis or the censor date (1 April 2014), whichever occurred first. The median follow-up was 16.9 years. The relationships between adipose indicators and weight change; and postmenopausal breast cancer were explored using Cox's proportional hazards regression. HRs with 95% CI were reported. Linear trends, using the exposure categories as a continuous variable were also presented. Potential confounders such as age, highest education levels achieved, time spent walking (hours/day), family history of breast cancer, oral contraceptive (OC) use, HRT use, parity, age at first birth, age of menarche and dietary pattern (meat eaters, fish eaters and vegetarians) were included in the fully adjusted models. In addition, BMI at 20 years was also adjusted for all adipose indicators, except BMI at 20 years. All statistical analyses were conducted using Stata V.14.

## RESULTS
After excluding the prevalent cases at baseline and incident cases developed within 1 year of questionnaire returned, 17 715 postmenopausal women were included in this analysis. From 282 277 person-years follow-up, there were 946 incident breast cancer cases with an incidence rate of 3.35 per 1000 women.

The mean (SD) age of the postmenopausal participants at enrolment was 58.8 (7.5) years. The majority were white (99.1%), about one-fifth had tertiary qualifications, 10% were smokers, 11% self-reported being on a vegetarian diet, 3% reported a family history of breast cancer, about 41% had ever taken HRT, 52% ever used

OC and 17% were nulliparous. The mean (SD) BMI at enrolment and at 20 years of age were 24.9 (4.3) kg/m$^2$ and 21.5 (2.9) kg/m$^2$, respectively (table 1). The correlation between skirt size and blouse size was very high (r=0.91, p<0.001) while the correlation between BMI and waist circumference was only moderate (r=0.65, p<0.001). Skirt size was highly correlated with waist circumference (r=0.76, p<0.0001) while blouse size was highly correlated with BMI (0.78, p<0.0001) (online supplementary table 1).

Incidence rates (per 1000 women) for postmenopausal breast cancer increased by BMI categories at enrolment, BMI categories at 20 years old and weight change (from 20 years old to enrolment age) (table 2). Obese women at recruitment had the highest incidence rates while women with the highest weight loss from age 20 had the lowest incidence rate of postmenopausal breast cancer.

After adjustment for potential confounders, adipose indicators at enrolment namely BMI, waist circumference and clothing sizes (blouse and skirt sizes) were associated with increased risk for postmenopausal breast cancer (table 3). Postmenopausal women who were overweight (HR: 1.34, 95% CI: 1.09 to 1.65) or obese (HR: 1.27, 95% CI: 0.91 to 1.76) had higher risks of breast cancer compared with those with normal weight at enrolment. There was no association observed for women in the underweight category. No association was observed for BMI at 20 years with postmenopausal breast cancer risk.

BMI and blouse size (which indicate general obesity), as well as waist circumference and skirt size (which indicate central obesity), had positive association with postmenopausal breast cancer. There was an increase of 3% in HR for every unit increase in BMI and 6% in HR for every 5 cm increase in waist circumference after adjustment for confounders. There was an additional 1% increase in HRs for both BMI and waist circumference after additionally adjusted for BMI at 20 years for both BMI and waist circumference. For every 2-unit increase in blouse and skirt sizes (as UK clothes size increased by 2 units), the HRs for blouse and skirt sizes were 1.08 (95% CI: 1.01 to 1.14) and 1.12 (95% CI: 1.06 to 1.19), respectively. The HRs increased to 1.10 (95% CI: 1.03 to 1.18) and 1.14 (95% CI: 1.06 to 1.22) for blouse and skirt sizes, respectively, after additional adjustment for BMI at 20 years of age.

Weight change over adulthood was associated with increased risk for postmenopausal breast cancer (HR: 1.02, 95% CI: 1.01 to 1.03). For every 1 kg weight change, there was a corresponding 2% risk increase after adjusting for confounders. Similar results were observed when weight change was additionally adjusted for BMI at 20 years (table 4). There was an increasing trend in associated risks with postmenopausal breast cancer (p for trend <0.001) when weight change was categorised into groups of weight loss, stable weight and increased weight, using stable weight as the reference category.

**Table 1** Descriptive characteristics of postmenopausal women

| | Total | n (%) |
|---|---|---|
| Age (mean±SD) years | 17686 | 58.8±7.5 |
| Highest education achieved | 15521 | |
| No education | | 3950 (25.5) |
| O-level | | 4682 (30.2) |
| A-level | | 3700 (23.8) |
| Degree | | 3189 (20.6) |
| White | 17149 | 16995 (99.1) |
| Food groups | 15602 | |
| Meat eaters | | 12175 (78.0) |
| Fish eaters | | 1700 (10.9) |
| Vegetarians | | 1727 (11.1) |
| Smokers | 17714 | 1758 (9.9) |
| Walking (mean±SD) (hours/day) | 15866 | 0.94±0.77 |
| Family history of breast cancer | 17715 | 539 (3.0) |
| Ever taken HRT | 17031 | 7045 (41.4) |
| Ever used OC | 17362 | 8997 (51.8) |
| Parity | 17292 | |
| 0 | | 2937 (16.9) |
| 1–2 | | 8685 (50.2) |
| 3–4 | | 5129 (29.7) |
| ≥5 | | 541 (3.1) |
| Age of menarche (mean±SD) (years) | 17247 | 12.9±1.6 |
| Age of first birth (mean±SD) (years) | 14336 | 25.3±4.4 |
| BMI at enrolment (mean±SD) (kg/m$^2$) | 17029 | 24.9±4.3 |
| Waist circumference (mean±SD) (cm) | 14491 | 75.3±9.7 |
| BMI at 20 years (mean±SD) (kg/m$^2$) | 16077 | 21.5±2.9 |
| Weight at 20 years (mean±SD) (kg) | 16388 | 57.5±8.3 |
| BMI categories at enrolment | 17029 | |
| Underweight (<18.5 kg/m$^2$) | | 327 (1.9) |
| Normal weight (18.5–24.9 kg/m$^2$) | | 9816 (57.6) |
| Overweight (25.0–29.9 kg/m$^2$) | | 4936 (29.0) |
| Obese (>30.0 kg/m$^2$) | | 1950 (11. 5) |
| BMI categories at 20 years old | 16077 | |
| Underweight (<18.5 kg/m$^2$) | | 1568 (9.8) |
| Normal weight (18.5–24.9 kg/m$^2$) | | 13128 (81.7) |
| Overweight (25.0–29.9 kg/m$^2$) | | 1176 (7.3) |
| Obese (>30.0 kg/m$^2$) | | 205 (1.3) |
| Skirt sizes | 16933 | |
| ≤10 | | 1036 (6.1) |
| 12 | | 3139 (18.5) |
| 14 | | 4806 (28.4) |
| 16 | | 4144 (24.5) |
| 18 | | 2238 (13.2) |
| ≥20 | | 1570 (9.3) |

Continued

**Table 1** Continued

|  | Total | n (%) |
|---|---|---|
| Blouse sizes | 16 248 |  |
| ≤10 |  | 1338 (8.2) |
| 12 |  | 3668 (22.6) |
| 14 |  | 4637 (28.5) |
| 16 |  | 3544 (21.8) |
| 18 |  | 1726 (10.6) |
| ≥20 |  | 1335 (8.2) |

BMI, body mass index; HRT, hormone replacement therapy; OC, oral contraceptive.

## DISCUSSION

Our results suggested that clothing size such as blouse and skirt sizes can be used as adipose indicators in predicting postmenopausal breast cancer. Trouser and skirt sizes have been reported to predict cancer risks independent of BMI,[10] while skirt size changes between 20 years of age, and postmenopausal age was reported to be associated with postmenopausal breast cancer.[11] However, blouse size has not been explored as a proxy for general obesity

**Table 2** Incidence rates of postmenopausal breast cancer by body mass index (BMI) groups and groups of weight change

|  | n | Person-years | Cases | Incidence rate (per 1000) |
|---|---|---|---|---|
| BMI groups at enrolment |  |  |  |  |
| Underweight | 327 | 5051 | 8 | 1.58 |
| Normal weight | 9813 | 158 418 | 475 | 2.99 |
| Overweight | 4936 | 77 987 | 300 | 3.85 |
| Obese | 1950 | 30 177 | 120 | 3.98 |
| Total | 17 026 | 271 634 | 903 | 3.32 |
| BMI groups at 20 years |  |  |  |  |
| Underweight | 1568 | 25 006 | 78 | 3.11 |
| Normal weight | 13 125 | 209 950 | 717 | 3.42 |
| Overweight | 1176 | 18 712 | 53 | 2.83 |
| Obese | 205 | 3165 | 10 | 3.16 |
| Total | 16 074 | 256 834 | 858 | 3.34 |
| Weight change |  |  |  |  |
| Weight loss ≥5 kg | 988 | 15 536 | 27 | 1.74 |
| Weight loss 2–4.9 kg | 745 | 12 116 | 25 | 2.06 |
| Stable weight (±1.9 kg) | 1820 | 29 436 | 87 | 2.96 |
| Weight gain 2–9.9 kg | 6287 | 101 530 | 336 | 3.31 |
| Weight gain 10–19.9 kg | 4450 | 70 498 | 264 | 3.74 |
| Weight gain ≥20 kg | 1969 | 30 632 | 125 | 4.08 |
| Total | 16 259 | 259 752 | 864 | 3.33 |

in predicting postmenopausal breast cancer. Our study confirms and provides additional evidence that skirt size is associated with the risk of postmenopausal breast cancer. In addition, our study may be the first to provide evidence that larger blouse size is associated with increased risk of postmenopausal breast cancer. The clothing sizes will provide additional indicators as proxy for adiposity if patients fail to recall (especially the elderly) or feel embarrassed to disclose their weight or BMI (for those who are obese).

Skirt size and waist circumference were highly correlated and performed equally well in predicting postmenopausal breast cancer risk. Our findings provide additional evidence of using skirt size as a proxy for central obesity in predicting postmenopausal breast cancer. On the other hand, there was no previous report on blouse size associated with increased risk of postmenopausal breast cancer. Blouse size could serve as an indicator for general obesity as it has a strong correlation with BMI. Blouse size also performed as well as BMI in predicting breast cancer among postmenopausal women. A previous study conducted in the UK reported that among 362 respondents, only 3% reported a clothing size different from their clothing labels.[9] This indicated that self-reported clothing size is reliable. Blouse and skirt sizes which are simple and non-invasive self-reported measures could be promoted as proxies for adipose indicators. From a public health perspective, these findings are significant when anthropometric measurements are not available, or challenging in their measurements due to extreme obesity.[10]

Overweight and obese women at enrolment had up to 32% increased risk of postmenopausal breast cancer. The association between overweight/obesity with postmenopausal breast cancer remained after further adjustment for BMI at 20 years. Women who gained the most weight since the age of 20 years old, had almost 30% increased risk of postmenopausal breast cancer after adjustment for potential confounders and BMI at 20 years. Similar findings have been reported by other researchers.[16–19]

Different periods of weight gain (ie, between age 18 years and the current age, between ages 18 and 35 years, between ages 35 and 50 years and between age 50 years and the current age) were consistently associated with increased breast cancer risk.[20] Similar findings were reported by Eliassen et al[21] that women who gained weight since age 18 years or since menopause were at an increased risk of breast cancer. These results suggest that regardless of BMI in early adulthood, being obese in later life or gaining weight along the way increases deposition of adipose tissue. After menopause, adipose tissue is the major location for the synthesis of oestrogens. Circulating oestrogen are higher in obese than in slim postmenopausal women.[19] Obesity also contributes to inflammatory changes leading to macrophage polarisation and altered adipokine profile.[22] These mechanisms may explain the association which links obesity and postmenopausal breast cancer together.

Consistent with previous literature[6 11 23]; waist circumference as an indicator for central obesity was found to be positively associated with postmenopausal breast cancer.

**Table 3** Crude and adjusted HR of general and central adiposity indicators

| | Age-adjusted HR (95% CI) | Model 1 HR (95% CI) | Model 2 HR (95% CI) |
|---|---|---|---|
| **BMI groups at 20 years** | | | |
| Cases/total | 857/16 055 | 457/8825 | NA |
| Underweight | 0.93 (0.73 to 1.15) | 0.97 (0.71 to 1.34) | – |
| Normal weight | 1.00 | 1.00 | |
| Overweight | 0.83 (0.63 to 1.01) | 0.81 (0.55 to 1.21) | – |
| Obese | 0.94 (0.51 to 1.76) | 0.42 (0.10 to 1.68) | – |
| Linear model (per kg/m$^2$) | 0.98 (0.96 to 1.00) | 0.98 (0.94 to 1.01) | – |
| P for trend (linear model) | 0.123 | 0.168 | – |
| **BMI groups at enrolment** | | | |
| Cases/total | 901/17 003 | 476/9241 | 453/8781 |
| Underweight | 0.54 (0.27 to 1.09) | 0.49 (0.16 to 1.55) | 0.32 (0.08 to 1.30) |
| Normal weight | 1.00 | 1.00 | 1.00 |
| Overweight | 1.27 (1.10 to 1.47) | 1.28 (1.05 to 1.56) | 1.34 (1.09 to 1.65) |
| Obese | 1.32 (1.08 to 1.62) | 1.18 (0.88 to 1.59) | 1.27 (0.91 to 1.76) |
| Linear model (per kg/m$^2$) | 1.03 (1.02 to 1.04) | 1.03 (1.00 to 1.05) | 1.04 (1.02 to 1.07) |
| P for trend (linear model) | | 0.021 | 0.001 |
| **Waist circumference at enrolment** | | | |
| Case/total | 776/14 476 | 483/9334 | 380/7377 |
| Waist circumference (per 5 cm) | 1.05 (1.02 to 1.09) | 1.06 (1.01 to 1.11) | 1.07 (1.01 to 1.14) |
| **Skirt size at enrolment** | | | |
| Case/total | 911/16 906 | 479/9197 | 447/8552 |
| ≤10 | 1.00 | 1.00 | 1.00 |
| 12 | 1.95 (1.29 to 2.96) | 2.06 (1.12 to 3.78) | 1.92 (1.04 to 3.54) |
| 14 | 2.02 (1.34 to 3.03) | 2.43 (1.35 to 4.39) | 2.45 (1.35 to 4.43) |
| 16 | 2.31 (1.54 to 3.48) | 2.74 (1.51 to 4.97) | 2.87 (1.58 to 5.22) |
| 18 | 2.32 (1.52 to 3.55) | 2.67 (1.43 to 4.98) | 2.74 (1.45.5.15) |
| ≥20 | 2.57 (1.66 to 3.97) | 3.21 (1.70 to 6.08) | 3.06 (1.58 to 5.93) |
| Linear model (per 2 unit) | 1.09 (1.05 to 1.15) | 1.12 (1.06 to 1.19) | 1.14 (1.06 to 1.22) |
| P for trend (linear model) | <0.001 | <0.001 | <0.001 |
| **Blouse size at enrolment** | | | |
| Case/total | 873/16 222 | 459/8834 | 428/8206 |
| ≤10 | 1.00 | 1.00 | 1.00 |
| 12 | 1.56 (1.13 to 2.17) | 2.04 (1.24 to 3.36) | 2.11 (1.26 to 3.53) |
| 14 | 1.59 (1.16 to 2.20) | 2.13 (1.30 to 3.48) | 2.25 (1.36 to 3.75) |
| 16 | 1.74 (1.25 to 2.42) | 2.20 (1.33 to 3.64) | 2.39 (1.42 to 4.03) |
| 18 | 1.57 (1.09 to 2.26) | 2.02 (1.16 to 3.52) | 2.21 (1.24 to 3.96) |
| ≥20 | 2.11 (1.46 to 3.03) | 2.63 (1.50 to 4.61) | 2.92 (1.60 to 5.32) |
| Linear model (per 2 unit) | 1.08 (1.03 to 1.12) | 1.08 (1.01 to 1.14) | 1.10 (1.03 to 1.18) |
| P for trend (linear model) | 0.001 | 0.019 | 0.005 |

Model 1: adjusted for age, education levels, parity, age at first birth, age of menarche, use of HRT, use of OC, family history of breast cancer, dietary pattern, walking (hours/day).
Model 2: model 1+BMI at 20 years.
BMI, body mass index; HRT, hormone replacement therapy; OC, oral contraceptive.

The association remained after adjustment for potential confounders and BMI at 20 years. This may indicate that regardless of being obese in early adulthood or at later life, being centrally obese in later life increases the risk of postmenopausal breast cancers. Central obesity is associated with increased insulin levels and insulin-like growth

**Table 4**  Crude and adjusted HR of weight change from age 20 years

| | Age-adjusted HR (95% CI) | Model 1 HR (95% CI) | Model 2 HR (95% CI) |
|---|---|---|---|
| **Weight change** | | | |
| Cases/total | 836/16 239 | 458/8921 | 453/8781 |
| *Weight loss* | | | |
| ≥5 kg | 0.59 (0.38 to 0.90) | 0.64 (0.34 to 1.19) | 0.59 (0.31 to 1.13) |
| 2–4.9 kg | 0.69 (0.44 to 1.08) | 0.67 (0.34 to 1.30) | 0.67 (0.34 to 1.29) |
| *Stable weight* | | | |
| ±1.9 kg | 1.00 | 1.00 | 1.00 |
| *Weight gain* | | | |
| 2–9.9 kg | 1.12 (0.88 to 1.41) | 1.23 (0.88 to 1.72) | 1.23 (0.88 to 1.72) |
| 10–19.9 kg | 1.25 (0.98 to 1.59) | 1.40 (0.99 to 1.98) | 1.41 (0.99 to 1.99) |
| ≥20 kg | 1.37 (1.04 to 1.80) | 1.30 (0.87 to 1.94) | 1.27 (0.85 to 1.91) |
| Linear model (per kg) | 1.02 (1.01 to 1.02) | 1.02 (1.01 to 1.03) | 1.02 (1.01 to 1.03) |
| P for trend (linear model) | <0.001 | <0.001 | <0.001 |

Model 1: adjusted for age, education levels, parity, age at first birth, age of menarche, use of HRT, use of OC, family history of breast cancer, dietary pattern, walking (hours/day).
Model 2: model 1+BMI at 20 years old.
BMI, body mass index; HRT, hormone replacement therapy; OC, oral contraceptive.

factors which may stimulate the growth of breast cancer cells and neoplastic degeneration.[24]

According to the latest report (2017) from the World Cancer Research Fund International/American Institute for Cancer Research, being overweight or obese during young adulthood decreases the risk of postmenopausal breast cancer.[6] Low levels of adiposity in the mammary gland may alter breast tissue maturation, making breast tissue more susceptible to carcinogenic stimuli among leaner women. Sex hormone levels may also explain the inverse relationship between early life adiposity and breast cancer risk. However, we did not find BMI at 20 years to be negatively associated with postmenopausal breast cancer incidence, as also reported by other cohort studies.[16 25]

There are a few limitations which need to be considered while interpreting the results. All adipose indicators, blouse and skirt sizes were self-reported. Some overweight or obese women may be more likely to under-report their weight compared with normal-weight participants. On the other hand, blouse and skirt sizes offer an objective standard as a proxy to report overweight/obesity which are potentially less liable for misreporting. We were unable to evaluate in more detail the effect of weight gain at different ages, other than that of weight at age 20 years and weight at enrolment. Breast tumour characteristics were not available, and we could not explore whether our findings may differ by breast cancer subtypes. On the other hand, we believe we are the first to report that both blouse and skirt sizes predict postmenopausal breast cancer risk equally as well as other more invasive measures of body size. In addition, our study sample is large with long follow-up duration. The linkage of participants'

ID with the National Statistics National Health Service central register provides accurate information on breast cancer diagnosis.

## CONCLUSION

Overweight/obesity (BMI and blouse size), central obesity (waist circumference and skirt size) and weight change during adulthood are associated with increased risk of postmenopausal breast cancer. Clothing size such as skirt and blouse sizes can be used as a proxy for adipose indicators in predicting postmenopausal breast cancer. Clothing sizes offer an objective standard as a proxy to report overweight/obesity which are useful in the public health setting. Maintaining healthy body weight over adulthood is an effective measure in the prevention of postmenopausal breast cancer.

**Acknowledgements**  We would like to acknowledge all women who participated in our study. The support from the Nutritional Epidemiology Group, University of Leeds is also acknowledged.

**Contributors**  Conception and design: FMM, JEC and DCG. Analysis and interpretation: FMM, JEC and DCG. Data collection: JEC and DCG. Manuscript preparation: FMM, JEC and DCG. Overall responsibility: FMM.

**Funding**  JEC was funded by a Medical Research Council grant no: MR/L02019X/1.

**Competing interests**  None declared.

**Patient consent**  Obtained.

**Ethics approval**  The National Research Ethics Committee for Yorkshire and the Humber, Leeds East.

**Provenance and peer review**  Not commissioned; externally peer reviewed.

**Data sharing statement**  The UK Women's Cohort Study is available for researchers to use at the following link: https://data.cdrc.ac.uk/dataset/uk-womens-cohort-questionnaire-data. The data are safeguarded and forms to apply

to access the data can be found here: https://www.cdrc.ac.uk/data-services/using-our-data/

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
