## [Reviewer comments · BMJ Open]

ARTICLE DETAILS

TITLE (PROVISIONAL)	Associations of clothing size, adiposity and weight change with risk of post-menopausal breast cancer in the UK Women's Cohort Study (UKWCS)
AUTHORS	Moy, Foong Ming; Greenwood, Darren; Cade, Janet

VERSION 1 – REVIEW

REVIEWER	Reiko Suzuki Japan women`s University, JAPAN
REVIEW RETURNED	08-Apr-2018

GENERAL COMMENTS	As long as I know this is first study to evaluate the clothing size and breast cancer risk, and the result of this manuscript is interesting. However, I think that clothing size could not always be an adipose indicator. People have certain personal preferences and style of the time. And I could not understand the following text in "Discussion" Page 8 line 4 to line 7, author wrote " Obese young women may be...(ref 24), which could render a protective effect for risk of breast cancer after menopause . However, we did not find BMI at 20 years to be positively associated with postmenopausal breast cancer incidence, as also reported by other cohort study (ref 16 23 25)." There are some inconsistencies in the above text. Further, there were few obese young women in the Japanese cohort (Ref 23), therefore this reference may be inappropriate in this sentence. I think the above text should be rewritten. Table 3. Liner model should not be used in trend test due to the possibility of U shape association.
--

REVIEWER	Cordina-Duverger Emilie INSERM UMRS1018, Center for research in Epidemiology and Population Health, Villejuif, France
REVIEW RETURNED	17-Apr-2018

GENERAL COMMENTS	In this manuscript, the authors investigated the association between various adipose indicators and post-menopausal breast cancer. This is a good paper in both form and substance. The major interest being for me the use of clothing size as adiposity indicators. The study design is clearly describe in a previous article, and the definition of post-menopausal women is clear and relevant. The selection of cases seems accurate and reliable, and adipose indicators are various and quite interesting.
---

	However, I have some observations. Materials and methods: 1) Page 5, line 21: It could have been more relevant to include in the model the HRT variable in 3 categories (current, past, never) since only current intake is associated with breast cancer. Results: 2) Page 5, line 38: the term "OCP" were not defined previously, why didn't you use the term OC? 3) Page 5, line 43: I think that it would be interesting to present the table with all the correlations (supplementary tables?) 4) Page 6, line 31: the authors have overstated this finding making appear as results of categories of weight change are statistically significant. Discussion: 5) Page 7, line 5: "Our results demonstrated " could be replaced by "our results suggested". 6) Page 7, line 36 : It could be interesting to enrich the discussion on adult weight change and the period of this weight change and add some references (Ahn, 2007 ; Eliassen, 2006...) 7) Page 8; line 8: Michel et al (2006) concluded that the decrease in breast cancer risk in obese women was not explained by menstrual cycle characteristics or ovulatory disorders. 8) Moreover, It was shown that the use of HRT could modulate the effect of adiposity on breast cancer risk. Have you performed some analyses which could support this hypothesis?
--	--

REVIEWER	Max Dieterich University of Rostock, Department of Obstetrics and Gynecology Breast Unit Gynecological Cancer Center, Germany
REVIEW RETURNED	14-Jun-2018

GENERAL COMMENTS	Comments on: „ Associations of clothing size, adiposity and weight change with risk of post-menopausal breast cancer in the UK Women's Cohort Study (UKWCS)" General comments: The authors are congratulated for conducting this large analysis of the UK Women's Cohort Study (UKWCS). Between 1995 and 1998, 17.781 women were recruited via a postal questionnaire. The aim was to show a clinical relation between clothing size and the risk of postmenopausal breast cancer. The main finding is, that increase blouse and skirt size can be used to predict post-menopausal breast cancer. From the reviewers point of view it is not surprising, that an increase in blouse size and skirt size is associated with an increased risk of postmenopausal breast cancer. An increase in clothing size implicates an increase in BMI. Increased BMI is positively associated with an increased risk for postmenopausal breast cancer. For the reviewer the clinical relevance is of this study remains unclear. A much easier way is the simple weight control using a scale. See M. L. Neuhouser et al. (Overweight, Obesity, and Postmenopausal Invasive Breast Cancer Risk A Secondary Analysis of the Women's Health Initiative Randomized Clinical Trials Marian L. Neuhouser et al. JAMA Oncol. 2015;1(5):611-621.): "Women with a baseline BMI of less than 25.0 who gained more than 5% of body weight over the follow-up period had an increased breast cancer risk (HR, 1.36; 95% CI, 1.1-1.65)" The manuscript is well written. All data are very well presented, and the Discussion is in line with the results. Overall the presented
--

	manuscript has a big sample size and no weakness in statistical or scientific context can be found. Abstract: no comments. Manuscript Introduction 1. What is the clinical benefit of measuring blouse size compared to weight control? Material and Methods: no comments Results: no comments Discussion: 2. What influence will these data have on clinical practice? Conclusion: The authors main finding is that blouse size can be predictive for postmenopausal breast cancer. No ground breaking new findings. However, a comparable study on skirt size and postmenopausal breast cancer was previously published in this journal (Fourkal et al. 2014, 4 (9): e005400) In conclusion: from a scientific perspective the paper is suitable for publication in BMJ Open. From a clinical perspective the manuscript is of minor interest.
--	---

VERSION 1 – AUTHOR RESPONSE

Reviewers' Comments to Author:

Reviewer: 1

Reviewer Name: Reiko Suzuki

Institution and Country: Japan women`s University, JAPAN

Competing Interests: None declared

As long as I know this is first study to evaluate the clothing size and breast cancer risk, and the result of this manuscript is interesting.

However, I think that clothing size could not always be an adipose indicator. People have certain personal preferences and style of the time.

Response: Thank you for the comments. We agree that personal preference and style influence the size of clothing and this will add some noise to the measurement. However, the clothing size will still generally follow one's body size, give or take some natural variation. We also note that weight and body mass index are not always indicative of adiposity, but may indicate greater muscle mass for example. Therefore, in the same way, we think clothing size can be used as an alternative proxy of adipose indicator, which may have some advantages in terms of recall.

And I could not understand the following text in "Discussion" Page 8 line 4 to line 7, author wrote

" Obese young women may be...(ref 24), which could render a protective effect for risk of breast cancer after menopause . However, we did not find BMI at 20 years to be positively associated with postmenopausal breast cancer incidence, as also reported by other cohort study (ref 16 23 25)."

Response: Have removed the sentence “Obese young women may be...(ref 24), which could render a protective effect for risk of breast cancer after menopause” to avoid confusion. (pg 13, line 237)

There are some inconsistencies in the above text. Further, there were few obese young women in the Japanese cohort (Ref 23), therefore this reference may be inappropriate in this sentence. I think the above text should be rewritten.

Response: We apologise for the inconsistencies. The text has been edited as “However, we did not find BMI at 20 years to be negatively associated with postmenopausal breast cancer incidence, as also reported by other cohort study (ref 16 23 25). We also removed the citation by Suzuki et al (ref 23) (pg13, lines 237-239)

Table 3. Linear model should not be used in trend test due to the possibility of U shape association.

Response: Thank you for the comment. However, from the WCRF continuous updates report which conducted dose-response meta-analyses of all existing cohorts investigating BMI and risk of breast cancer, it was concluded that “Breast cancer risk increased monotonically through all ranges of BMI” and in particular that “Postmenopausal breast cancer risk increased monotonically through all ranges of BMI” (reference below). The a priori evidence was therefore that it was unlikely there would be a nonlinear association with clothing size.

Reference - World Cancer Research Fund / American Institute for Cancer Research. Diet, Nutrition, Physical activity and Cancer: a global perspective. Continuous Update Project Expert Report 2018. Available at dietandcancer.org

Reviewer: 2

Reviewer Name: Cordina-Duverger Emilie

Institution and Country: INSERM UMRS1018, Center for research in Epidemiology and Population Health, Villejuif, France

Competing Interests: None declared

In this manuscript, the authors investigated the association between various adipose indicators and post-menopausal breast cancer. This is a good paper in both form and substance. The major interest being for me the use of clothing size as adiposity indicators.

The study design is clearly describe in a previous article, and the definition of post-menopausal women is clear and relevant. The selection of cases seems accurate and reliable, and adipose indicators are various and quite interesting.

However, I have some observations.

Materials and methods:

1) Page 5, line 21: It could have been more relevant to include in the model the HRT variable in 3 categories (current, past, never) since only current intake is associated with breast cancer.

Response: We agree with your suggestion. Unfortunately, the data on oral contraceptive (OC) use was collected as yes/no to ever use of OC.

Results:

2) Page 5, line 38: the term "OCP" were not defined previously, why didn't you use the term OC?

Response: Have edited as "...oral contraceptive (OC) use..." (pg 5, line 120) and edited accordingly in Table 1 and footnotes of Tables 3-5.

3) Page 5, line 43: I think that it would be interesting to present the table with all the correlations (supplementary tables?)

Thank you for the suggestion. We added the following table as "Supplementary table"

Correlations between Skirt size, blouse size, waist circumference and BMI

	Skirt size	Blouse size	Waist Circumference	BMI
Skirt size	1.00			
Blouse size	0.91	1.00		
Waist circumference	0.76	0.73	1.00	
BMI	0.75	0.78	0.65	1.00

All p values <0.001

4) Page 6, line 31: the authors have overstated this finding making appear as results of categories of weight change are statistically significant.

Response: We edited the sentence as "There was an increasing trend in associated risks with post-menopausal breast cancer (P for trend <0.001) when weight change was categorised into groups of weight loss, stable weight and increased weight, using stable weight as the reference category" and removed "... women who experienced weight loss had lower risk, while those with weight gain had higher risk for post-menopausal breast cancer..." (pg 11, lines 179-181)

Discussion:

5) Page 7, line 5: "Our results demonstrated " could be replaced by "our results suggested".

Response: Edited accordingly (pg 12,line 190)

6) Page 7, line 36 : It could be interesting to enrich the discussion on adult weight change and the period of this weight change and add some references (Ahn, 2007 ; Eliassen, 2006...)

Response: Added the following text :

Different periods of weight gain (ie: between age 18 years and the current age, between ages 18 and 35 years, between ages 35 and 50 years, and between age 50 years and the current age) were consistently associated with increased breast cancer risk [20]. Similar findings were reported by Eliassen et al [21] that women who gained weight since age 18 years or since menopause were at an increased risk of breast cancer. (pg 13,lines 217-221)

7) Page 8; line 8: Michel et al (2006) concluded that the decrease in breast cancer risk in obese women was not explained by menstrual cycle characteristics or ovulatory disorders.

Response: Thank you for the suggestion. Welti et al, 2017 [19] reported that “Circulating oestrogen are higher in obese than in slim post-menopausal women”. The findings were reported from an established cohort study (Women Health Initiative) with more than 80,000 women followed up over 20 years, which we think the evidence is strong and recent. We are sorry that we could not find the study by Michel et al (2006) to make comparison. In addition, we think the findings by Michell et al were outdated.

8) Moreover, It was shown that the use of HRT could modulate the effect of adiposity on breast cancer risk. Have you performed some analyses which could support this hypothesis?

Response: Thank you for the suggestion. We didn't perform the analysis as this was not our main research question.

Reviewer: 3

Reviewer Name: Max Dieterich

Institution and Country: University of Rostock, Department of Obstetrics and Gynecology Breast Unit Gynecological Cancer Center, Germany

Competing Interests: None declared

General comments:

The authors are congratulated for conducting this large analysis of the UK Women's Cohort Study (UKWCS). Between 1995 and 1998, 17.781 women were recruited via a postal questionnaire. The aim was to show a clinical relation between clothing size and the risk of postmenopausal breast cancer.

The main finding is, that increase blouse and skirt size can be used to predict post-menopausal breast cancer.

From the reviewers point of view it is not surprising, that an increase in blouse size and skirt size is associated with an increased risk of postmenopausal breast cancer. An increase in clothing size implicates an increase in BMI. Increased BMI is positively associated with an increased risk for postmenopausal breast cancer. For the reviewer the clinical relevance is of this study remains unclear.

A much easier way is the simple weight control using a scale. See M. L. Neuhouser et al. (Overweight, Obesity, and Postmenopausal Invasive Breast Cancer Risk A Secondary Analysis of the Women's Health Initiative Randomized Clinical Trials Marian L. Neuhouser et al. JAMA Oncol. 2015;1(5):611-621.): "Women with a baseline BMI of less than 25.0 who gained more than 5% of body weight over the follow-up period had an increased breast cancer risk (HR, 1.36; 95% CI, 1.1-1.65)"

Response: We agree that weight is a simple adipose indicator. However, this requires a weighing scale or some participants may not have measured their weight or unable to recall their weight. Clothing size may be easier to recall when women have purchased clothes using the standard clothing sizes as happens in the UK.

The manuscript is well written. All data are very well presented, and the Discussion is in line with the results. Overall the presented manuscript has a big sample size and no weakness in statistical or scientific context can be found.

Abstract: no comments.

Manuscript

Introduction

1. What is the clinical benefit of measuring blouse size compared to weight control?

Response: Clothing sizes offer an objective standard as a proxy to report overweight / obesity, which are potentially less liable for misreporting. Weight control requires a weighing scales which are subjected to measurement bias and recall bias. From a public health perspective, these findings are significant when anthropometric measurements are not available, or challenging in their measurements due to extreme obesity.

Added the following text:

Clothing sizes are objective, easy to recall and less likely to be misreported. From a public health perspective, clothing sizes may be useful when anthropometric measurements are not available or challenging in their measurements due to extreme obesity. (pg 3, lines 60-62)

Material and Methods: no comments

Results: no comments

Discussion:

2. What influence will these data have on clinical practice?

Response: Added in Discussion : The clothing size will provide an additional indicator as proxy for adiposity when anthropometric measurements are not available, or challenging in their measurements due to extreme obesity, if patients can't recall (especially the elderly) or feel embarrassed to disclose their weight or BMI (for those who are obese). (pg 12, lines 197-200)

Conclusion:

The authors main finding is that blouse size can be predictive for postmenopausal breast cancer. No ground breaking new findings. However, a comparable study on skirt size and postmenopausal breast cancer was previously published in this journal (Fourkal et al. 2014, 4 (9): e005400)

Response: Our results confirm the findings of Fourkala et al, which only reported on the relationship between skirt size and breast cancer risk, and provides additional evidence that besides skirt size, blouse size is also associated with postmenopausal breast cancer risk.

In conclusion: from a scientific perspective the paper is suitable for publication in BMJ Open. From a clinical perspective the manuscript is of minor interest.

Response: We consider that using clothing size as an additional adipose indicator linked to breast cancer risk will be useful in the public health setting providing easily accessible information which does not require accurate weighing scales to be available.

Added the following text: Clothing sizes offer an objective standard as a proxy to report overweight / obesity, which are useful in the public health setting. (pg 14, lines 256-257)

FORMATTING AMENDMENTS (if any)

Required amendments will be listed here; please include these changes in your revised version:

- Kindly remove the tables uploaded separately and place it inside your main document where it was first cited.

Response: Edited accordingly

- Please embed your DATA SHARING STATEMENT in your main document file.

Response: We have added the following :

The UK Women's Cohort Study is available for researchers to use at the following link: <https://data.cdrc.ac.uk/dataset/uk-womens-cohort-questionnaire-data>. The data is safeguarded and forms to apply to access the data can be found here: <https://www.cdrc.ac.uk/data-services/using-our-data/> (pg 15, lines 276-280)

- Patient and Public Involvement:

Authors must include a statement in the methods section of the manuscript under the sub-heading 'Patient and Public Involvement'.

This should provide a brief response to the following questions:

How was the development of the research question and outcome measures informed by patients' priorities, experience, and preferences?

How did you involve patients in the design of this study?

Were patients involved in the recruitment to and conduct of the study?

How will the results be disseminated to study participants?

For randomised controlled trials, was the burden of the intervention assessed by patients themselves?

Patient advisers should also be thanked in the contributorship statement/acknowledgements.

If patients and or public were not involved please state this.

Response: We have added the following :

The participants were not involved in the development of the research question and outcome measures. They were also not involved in the design and conduct of the study. The results are not disseminated to the participants apart from the UKWCS website. (pg 4,lines 89-92)

VERSION 2 – REVIEW

REVIEWER	Max Dieterich University of Rostock, Department of Obstetrics and Gynecology Breast Unit Gynecological Cancer Center, Germany
REVIEW RETURNED	27-Jul-2018
GENERAL COMMENTS	Comments on the revised manuscript: „ Associations of clothing size, adiposity and weight change with risk of post-menopausal breast cancer in the UK Women’s Cohort Study (UKWCS)” General comments: The prior concerns of the reviewer were addressed and answered to the reviewer’s satisfaction. Still, the clinical relevance remains controversial. Obese patients are identified by clinical judgement and it is irrelevant if a patient weights 100kg or 130kg. Within these “extreme obese” patients the risk of breast cancer is increased. Patients might be embarrassed as well in communicating their clothing size. However, the study is scientific sound. The Editor has to decide whether this manuscript fits into the context of BMJ Open.